# The Skin Microbiome and Influencing Elements in Cutaneous T-Cell Lymphomas

**DOI:** 10.3390/cancers14051324

**Published:** 2022-03-04

**Authors:** Marion Jost, Ulrike Wehkamp

**Affiliations:** Department of Dermatology, Venerology and Allergology, University Hospital Schleswig-Holstein, Campus Kiel, 24105 Kiel, Germany; uwehkamp@dermatology.uni-kiel.de

**Keywords:** microbiome, cutaneous T-cell lymphoma, staphylococcus aureus, antimicrobial peptides, skin barrier, antibiotic treatment, immunology, filaggrin, review

## Abstract

**Simple Summary:**

Since the 1970s, a connection between microbes living on the skin and the rare cutaneous neoplasia, cutaneous T-cell lymphomas (CTCL), was suggested. New technologies, for instance, next-generation sequencing technologies, enable investigators to look closely at the interplay between microbes and the host. In the present review, we collected research regarding the role of skin microbiota and skin barrier elements in the most common CTCL. It is known that Staphylococcus aureus infections play a major role in morbidity and mortality in advanced stages of the disease. It is possible that the microbiota of the patient might be involved in disease progression or its origin. Some findings suggest that the skin barrier may be deficient in CTCL. Restoring the skin barrier in CTCL might be a promising therapeutical option. Further studies are needed to provide more insight and potentially contribute to the development of new treatment approaches.

**Abstract:**

Since the 1970s, a connection between the skin’s microbiota and cutaneous T-cell lymphomas (CTCL) was suggested. New techniques such as next-generation sequencing technologies enable the examination of the nuanced interplay between microbes and their host. The purpose of this review is an updated description of the current knowledge on the composition of the microbiome, relevant bacteria, or other stimuli, and their potential role in CTCL with a focus on the most frequent subtype, mycosis fungoides. Some findings suggest that the skin barrier—or the deficiency hereof—and host-microbiota might be involved in disease progression or etiopathogenesis. In addition, information on the current knowledge of antimicrobial peptide expression in CTCL, as well as treatment considerations with antiseptics and antibiotics, are included. Further studies are needed to provide more insight and potentially contribute to the development of new treatment approaches.

## 1. Introduction

### 1.1. Skin Microbiome

The human skin represents the largest human organ. It provides an effective barrier between the human organism and the environment. Superficial skin layers are inhabited by different sorts of microorganisms, such as bacteria, viruses, archaea, and fungi [1]. This heterogeneous community of microorganisms are in mutualistic symbiosis. They play an essential role in the protection against invading pathogens and in the breakdown of natural products. Additionally, they contribute to a special form of innate and adaptive immunity, which links antimicrobial functions and tissue repair [2]. They are able to modulate the production of antimicrobial peptides (AMPs) and various cytokines in and on the skin (e.g., IL-1, IL-17, IL-22, TNF) [1,3]. The genome as a whole, encompassing all of the mentioned microorganisms, is known as the skin microbiome. 

Some bacteria are part of the common skin microbiota, e.g., *Staphylococcus epidermidis (SE)*, and ubiquitously colonize human skin, as opposed to others, e.g., *Staphylococcus aureus (SA)*, which are rarely found on healthy human skin [4].

Under normal conditions, transient skin microbes can persist on the skin only for a few hours or days and are not pathogenic (normal immune responses, skin barrier function intact) [1]. If the skin barrier is deficient, or if the microbial balance between commensals and pathogens is disturbed, skin diseases or even systemic diseases can be triggered [2]. Furthermore, it is known that in a chronic inflammatory environment, malignant T-cells are prone to proliferate [5].

### 1.2. Cutaneous T-Cell Lymphomas

Primary cutaneous T-cell lymphomas (CTCL) represent a group of non-Hodgkin lymphomas characterized by the accumulation of clonally expanded CD4+ T-cells in the skin. Mycosis fungoides (MF) is the most common subtype and represents around 65–75% of these cases. The etiopathogenesis is still unknown, and a curative treatment option is not yet available. However, disease control can be achieved through stage-adapted therapy, especially in early stage patients [6,7].

CTCL, especially early stages of MF and SS, often resemble chronic inflammatory dermatoses. In particular, the clinical picture is often misinterpreted as eczema or atopic eczema. This often results in a delayed diagnosis for the patient. The incidence of CTCL is currently around 0.4–0.6/100,000/year [7].

The skin lesions in MF are described as erythematous patches and plaques, which can ulcerate, and in advanced stages, patients can develop tumors. SS is an aggressive variant of CTCL that is clinically associated with erythroderma, lymphadenopathy, and leukemia [8,9]. A cure of the disease is currently not possible, but through consistent and stage-adapted therapy, disease control can be achieved. Topical glucocorticosteroids, retinoids, UV therapies, immunosuppressants, radiation therapies, and various immunomodulatory treatments are used for treatment [6,10,11,12,13,14,15,16,17,18,19].

Since the clinical skin changes in CTCL may mimic inflammatory skin disorders, the question was addressed, whether the skin microbiota might play a significant role in the development or in the progression of the disease. In the 1960s and 1970s, the first case reports about the positive effects of antibiotics and antiviral therapy in CTCL were described. Since then, it was suspected that a chronic antigen stimulus might trigger the development and/or the progression of the disease (Figure 1) [20].

In inflammatory skin diseases, both changes in the expression of antimicrobial peptides on the skin surface and changes in microbial colonization are known. Cancer-induced skin barrier defects and potential shifts in the skin microbiome might render CTCL patients prone to increased susceptibility to bacterial infections [16]. This could be of special relevance for CTCL patients since it is common knowledge that *SA* infections play a major role in morbidity and mortality in these patients [22,23].

A PubMed search (Appendix A) was conducted for studies evaluating the skin microbiota and other influencing elements in the most common cutaneous T-cell lymphomas, MF, and Sézary Syndrome (SS), and elaborated in a narrative way.

## 2. Microbiome Analysis

The role of the microbiota in CTCL is not well understood yet. The recent establishment of next-generation sequencing methods and reference databases puts the skin microbiome and its role in the pathogenesis of CTCL in the spotlight.

Very recently, two studies on the skin microbiome in CTCL were published. These studies are focused on MF, and a further one on parapsoriasis, respectively. The literature was also checked for data on rarer subtypes of CTCL, e.g., CD4+ small/medium-sized lymphoproliferation and primary cutaneous CD30+ lymphoproliferative disorders. However, there is no evidence published about the microbiome in these entities.

Harkins et al. analyzed skin swabs from four MF patients (stages IA to IIIA) and two SS patients (stage IVA1), matched with samples from age- and sex-matched healthy volunteers. Via “shotgun metagenomic sequencing”, only slight shifts in the skin microbiota were noticed. They observed increasing trends in the mean relative abundances of *Corynebacterium species* and decreasing trends in *Cutibacterium species* without statistical significance. The authors suggested that the bacterial shifts may correlate with disease stage or treatment status [21].

Another group, Salava et al., analyzed skin swabs from 20 Patients with MF at stages IA-IIB. They matched their data from both 16S rRNA gene sequencing” and “whole-genome shotgun sequencing” to swabs from contralateral healthy-appearing body sites on the same patients. This group also could not detect significant differences at the genus level or in the microbial diversity in the composition of the skin microbiome in their analyzed patients [24].

In 2017 the same group already looked into microbiome changes in parapsoriasis-affected skin. In comparison to healthy-appearing, contra-lateral skin, there were not any statistically significant differences detectable [25]. Parapsoriasis is primarily considered an inflammatory disease. However, it was discussed that parapsoriasis might represent a precursor to the development of lymphoma. Clinically, patients present persistent, finely scaling, and mildly eczematic lesions that might resemble early stage MF [26].

In summary, the CTCL skin microbiome analyses did not yield statistically significant results, probably due to the small number of patients. CTCL is a rare disease; hence, multicenter analyses, the inclusion of larger patient numbers and investigations according to the same study protocol, should be considered to find statistically significant differences in the future.

### 2.1. Location Sites

Human skin sites provide diverse microenvironments that vary in pH, temperature, moisture, sebum content, and ultraviolet light exposure. Due to these characteristics, the sites can be grouped in sebaceous (face, chest and back), moist (bend of elbow, back of knee and groin), and dry (volar forearm and palm). Dependent on the physiology of the skin site, the composition of microbial communities changes in the relative abundance of bacterial taxa [2,27,28]. Despite constant environmental change, skin microbial communities are quite stable at least over a two-year time period [29].

### 2.2. Methods

The skins’ microorganisms are analyzed either by “amplicon sequencing” or “shotgun metagenomics”. In “amplicon sequencing”, the unique “16S ribosomal RNA gene” is sequenced for bacteria, which is called “16S rRNA gene sequencing/analysis”. For fungi, the internal transcribed spacer 1 (ITS1) region of the eukaryotic ribosomal gene is used. This follows assembly or mapping to a reference database. The “16S rRNA gene sequence” is like a unique “barcode” for every microbe. In contrast, “shotgun metagenomics/whole genome sequencing” captures the entire complement of genetic material in a sample without a previous targeted amplification step, either for DNA or RNA, which also includes the hosts’ genetic information [2,27]. An overview of the methods that were applied to CTCL/parapsoriasis samples in recent studies is given in Table 1.

### 2.3. Healthy Controls

Matching the results from the microbiome analysis with the contralateral healthy-appearing skin or with the results from a healthy volunteer is essential because the ecological body site is a greater determinant of the microbiota composition than individual genetic variation. This means that the antecubital fossa, back, and plantar heel are more similar to the same site on another individual than to any other site on the same individual [1,27,30]. This knowledge is important for any kind of microbiome analysis and not specific for CTCL patients.

### 2.4. General Limitations of Skin Microbiome Analyses

It is important to note that there are still limitations in the current microbiome analyses performed by “amplicon sequencing” or “whole-genome metagenomis”. Both cannot differentiate between living and dead microorganisms. To resolve this issue, it might be feasible to pre-digest and remove dead microbial cells from the analysis, to obtain a more accurate assessment of the living microbiome [28].

Skin microbiome analysis usually relies on skin swabs; however, some microorganisms are variably present at the surface compared with deeper skin layers. These issues need to be addressed in future study protocols and are the topic of a recent review by Byrd et al. [2].

## 3. Bacteria

*SA* and *SE*, both members of the Staphylococcus genus, represent the main players when talking about skin microbiota. In CTCL, *SA* infections are known to play a major role in morbidity and mortality in these patients. However, the question of if the abundance of *SA*, its toxins, or the cancer-induced skin barrier defects are the major problem is not yet clarified [22,23,31].

### 3.1. Staphylococcus aureus and Staphylococcus aureus Enterotoxin

*SA* can generate a pro-oncogenic milieu in lesional skin in vivo. Supporting the hypothesis of a potential relevance of *SA*, CD4+ T-cell responses to *SA* can inadvertently enhance neoplastic progression in models of CTCL [22,32]. 

Staphylococcal enterotoxins (SEs) released by certain lineages of *SA* act as a class of “superantigens” that are extremely potent activators of T-cells [13,33,34].

The toxins provide access to deeper layers of the skin by disrupting the cell–cell contacts between keratinocytes [4]. The enterotoxins bind directly as whole proteins to major histocompatibility complex class II molecules outside the antigen-peptide binding groove and to certain families of T-cell receptor (TCR) V beta chains crosslinking TCR complexes and inducing T-cell activation at extremely low concentrations [22]. Willerslev-Olsen et al. suggested that toxin-mediated activation of malignant cells do not rely on the expression of a single, toxin-specific TCR-Vb chain in malignant cells but on the expression of multiple toxin binding, TCR-Vb chains expressed in bystander non-malignant tumor-infiltrating T-cells [13]. 

In CTCL patients, an increased prevalence of human leukocyte antigen (HLA)-DR5 (DR11) and HLA-DQ*03, and in SS HLA-DQ*0502 alleles (odd’s ratio = 7.75) were observed. HLA class II antigens affect the binding of bacterial and present processed antigen to CD4+ T-cells [35]. This may suggest that MF and SS occur in genetically determined hosts [30]. 

The presence of alpha-toxin further favors the persistence of malignant cells while removing the non-malignant CD4+ T-cells [36]. In 2021, Willerslev-Olsen et al. found evidence that *SA* and its toxins (SEs) induce the expression of oncogenic miR-155 in CTCL. Furthermore, they were able to show that in two patients with SS, aggressive antibiotic therapy was associated with decreased miR-155 expression in lesional skin [31]. SEs can also activate signal transducer and activator of transcription 3 (STAT3) and induce the expression of IL-17 [13]. Krejsgaard et al. wrote “that it seems plausible that IL-17 indirectly influences CTCL tumorigenesis by modulating angiogenesis and inflammation” [37].

Lindahl et al. summarized in 2021, that via *SA* toxins triggered effects: “(i) expression of oncogenic miR-155 and regulatory proteins (PD1, FoxP3, and IL-10), (ii) STAT3 and STAT5 activation, (iii) inhibition of anti-tumor cytotoxicity, and (iv) proliferation of primary malignant T-cells in vitro” [33]. According to these results, *SA* and its toxins might become a therapeutic target in CTCL [36].

### 3.2. Staphylococcus epidermidis (SE)

Several mechanisms were characterized to describe the effects that *SE* displays with regard to the skin and the skin barrier.

*SE* produces lantibiotics (bacteriocins), which are lanthionine containing antibacterial peptides. These induced peptides enhance the skin response via pattern recognition receptors (PRRs), contributing to the initial sensing of microorganisms and intracellular signal transduction [3]. Through the activation of Toll-like receptor 2 (TLR-2), the production of proinflammatory cytokines via TLR-3 is downregulated, and the expression of tight junctions is upregulated [1]. This indicates that innate immune sensing of microbial products does not only involve one TLR ligand but rather a whole pattern of potential ligands [4].

Interestingly, *SE* can drive skin injury through the secretion of the cysteine protease (EcpA). Its exact role in CTCL needs to be evaluated in the future [20].

### 3.3. Other Bacteria

The microbiome consists of a large number of different bacteria. In this paragraph, we summarize findings that were published beyond *SA* and *SE* in CTCL. All of these bacteria were only mentioned in single studies. Accordingly, the findings have to be interpreted with the appropriate caution.

In a recent microbiome analysis, Harkins et al. saw increasing trends in the mean relative abundances of *Corynebacterium species* and decreasing trends in *Cutibacterium species*. They suggested a possible correlation to the disease stage [21].

Another group reported 10 MF patients with dark brown to black necrotic tumors with eschars [34]. These tumors had *Enterococcus* cultured from a swab or tissue culture and healed or resolved entirely under appropriate antibiotic therapy [30].

*Pseudomonas aeruginosa* is considered a factor for a fatal outcome in septic CTCL patients. Together with *SA*, *Pseudomonas aeruginosa* is acknowledged to be associated with more than 50% of deaths in this patient group [38]. Whether these bacterial colonizations are primary or secondary findings is not yet clear.

## 4. Antimicrobial Peptides and Filaggrin

Antimicrobial peptides (AMP) are mainly produced in the skin by keratinocytes, neutrophils, and mast cells. These peptides act against bacteria, mycobacteria, enveloped viruses, and fungi [16,39]. Inflammatory or mechanical triggers induce their expression on the skin [40].

Gambichler et al. showed an upregulation of two AMP human-beta-Defensine-2 (hBD-2) and hBD-3 comparing healthy skin, lesional, and non-lesional tissue in 13 MF patients. hBD-1 was downregulated [39].

In 26 CTCL patients examined by Suga et al., psoriasin (S100A7) was low in lesional CTCL skin compared with psoriatic skin samples [16]. However, the analyses of the expression levels of psoriasin in lesional CTCL skin compared to non-lesional skin of the same patient yielded a higher expression in lesional skin areas [40]. Similarly, lesional plaque and tumor skin of CTCL expressed significantly higher levels of S100A8 mRNA and S100A9 mRNA in comparison to non-lesional skin [16]. Further, Nakajima et al. reported a decreased expression of Progranulin (PGRN) in both serum and lesional skin of MF patients [41].

The dysregulation of the AMP might reflect a regular response to bacterial colonization. Alternatively, the change in the AMP expression could be due to the CTCL itself, potentially producing the inflammation and a subsequently dysregulated signaling [5].

In addition to AMPs, other factors are known to be relevant for maintaining the physiological function of the skin barrier. One of them is filaggrin, which is extensively studied in inflammatory skin diseases, namely atopic dermatitis [42]. Filaggrin is an epidermal protein that aggregates keratin filaments and provides a cytoskeleton for the cornified envelope. Filaggrin mutations lead to increased surface pH and altered expression of AMP [1].

Suga et al. examined the mRNA expression levels of filaggrin in CTCL skin. CTCL skin contained lower levels of filaggrin compared to normal skin. The expression levels of filaggrin correlated negatively with serum levels of disease markers [16]. Trzeciak et al. mentioned gene expression of filaggrin (FLG, FLG2), repetin (RPTN), and small proline-rich protein (SPRR1A) as potentially key players in CTCL skin barrier deficiency. In their study from 2020, the levels of the transcripts of genes encoding these cornified envelope (CE) proteins differed significantly in comparison to healthy controls [42]. These findings strongly indicate skin barrier deficiency in CTCL.

## 5. Antiseptic and Antibiotic Treatment

The treatment of CTCL with antiseptics and antibiotics is of interest, as the fatality of infections in advanced-stage patients was well-known for more than fifty years [20]. A summary of the different previously published treatment approaches is given in Table 2.

In 2008, Nguyen et al. revealed that *SA* colonization appears to correlate directly with the body surface area of CTCL. Therefore, in order to decrease the severity of the disease, patients with extensive skin involvement were bathed in diluted sodium hypochlorite (one-quarter cup of 6% sodium hypochlorite in a full bathtub of water). The frequency of baths ranged from once daily to once weekly, depending on the severity of symptoms [23].

In 2018, the “Duvic regimen”, combining aggressive antibiotics (vancomycin and cefepime), an antiseptic whirlpool bathing system (with chlorhexidine gluconate 0.4%), and steroids (0.1% Triamcinolone ointment, respectively, 1.0–2.5% hydrocortisone cream), was applied under wet wraps and Mupirocin 2% ointment alternating with 1% silver sulfadiazine cream on open ulcers, followed by skin moisturization as described. This treatment regimen had a beneficial effect in a CTCL patient with erythroderma and methicillin-resistant *SA* [43].

In 2019, Lindahl et al. performed a study to test combinational antibiotics in eight patients with advanced-stage CTCL, regularly monitoring the microbiome with skin swabs and characterization of the T-cell populations in the skin biopsies. The patients were treated for 24 days with antibiotics (Cephalosporin and metronidazole D1–10, a combination of amoxicillin, and clavulanate D11–24).

In five of six patients, the frequency of the most dominant TCR clonotype decreased significantly (*p* > 0.05) 60 days after the initiation of antibiotic therapy [22]. Concurrent with the decrease in the presumably malignant dominant T-cell clone, a relative increase in the less frequent presumably non-malignant T-cell clones were seen. The relative distribution between the TCR-Vb families remained largely unchanged [22].

Interestingly, one patient continued to improve clinically after the termination of the antibiotic treatment regime, although *SA* reemerged on the skin. This result suggests that antibiotics may lead to a lasting change in the tumor and its microenvironment [22].

Very recently, in 2021, El Sayed et al. reported a decrease in the disease activity scores under doxycycline therapy in CTCL patients. In this study, the decrease was inferior to psoralen plus UV-A (photochemotherapy) treatment [44]. Besides its antibiotic value, doxycyclines capacity to inhibit *nuclear factor-kB* is under observation in these studies [45]. Alterations in the *nuclear factor-kB* pathway in malignant T-cells was shown previously [15].

Whether the positive effect of antibiotics on the malignant CTCL cells is mediated directly or potentially indirectly is still up for debate. Indirect effects could be mediated through the inhibition of bystander T-cells and the induction of anti-inflammatory effects. Lindahl and colleagues observed that high-affinity receptors for IL-2, STAT3 signaling, and proliferation were inhibited in lesional skin following antibiotic treatment [22,44].

## 6. Conclusions

In the 1980s, Shelley et al. and others suggested that chronic antigenic stimulation of the T-cell via chronic infections—as a source of continuous antigen release—may result in MF [20]. Another group summarized that chronic antigen exposure through microbial components or superantigens from bacteria such as *Staphylococcus* might result in the development of CTCL in genetically determined hosts [30].

The role of T-cell immune function in controlling disease progression can be seen in a decreased complexity of the T-cell-receptor repertoire in advanced CTCL, which is comparable to AIDS, illustrating the compromised growth of normal T-cells in both conditions [45]. A shift in the composition of infiltrating T-cells can be observed during the course of the disease. In the early stages, the infiltrate consists of non-malignant TH1-cells and cytotoxic CD8+ cells, in contrast to later stages with a predominance of TH2-cells and cytokines. It is suggested that the malignant T-cells are most likely responsible for this TH1/TH2 shift [32].

Microbial products and bacterial infections may stimulate disease progression in CTCL because both, strong activation of STAT-3 and high expression of IL-10, were identified as markers of poor response to treatment in CTCL [30]. Interestingly, in an under germ-free conditions living mouse model of CTCL, no disease progression was observed [15]. 

After antibiotic therapy, it was shown that the frequency of the most dominant T-cell-receptor clonotype decreases significantly [22]. Nevertheless, the simple long-term use of antibiotics and antiseptics might not be the ultimate solution due to the selection pressure of the bacteria and possible resistance mechanisms. In early stage disease, skin-directed treatments are in many cases sufficient, as opposed to advanced-stage disease. For the latter, antibiotic treatment, and other antibacterial measures, might pave the way for a better subsequent or concurrent response to CTCL-directed therapies [22,46].

Krejsgaard et al. proposed “that the most effective treatment of progressive and advanced disease should rationally combine therapeutics that directly target the malignant T-cells with drugs that (i) enhance cellular immunity, (ii) neutralize immune evasive mechanisms, (iii) inhibit the pro-tumorigenic environment, and (iv) eliminate pro-oncogenic bacteria such as enterotoxin- producing *SA* in infected patients” [5].

Changes in the skin microbiome could contribute to both the pathogenesis of CTCL and the maintenance or progression of the disease. The interaction between the skin microbiota, the structural cells of the skin, and the immune system is complex and not yet fully understood. The majority of CTCL patients have a benign, slowly progressive course; however, some patients rapidly develop tumors, and ultimately die of lymphoma or the complications caused by it, such as infections and sepsis.

Cancer-induced skin barrier defects could play an important role in the increased susceptibility to bacterial infections [22,23]. Skin barrier deficiency contributes to the exacerbation of immunological processes due to skin infections [42]. An important step in the treatment of CTCL might be the reinforcement of the skin barrier integrity via a balanced skin microbiome (restoration of diversity in the skin microbiome) and a reduction of inflammatory stimuli. Emollients that promote the skin barrier integrity might be beneficial for CTCL patients.

Further studies are necessary to describe and understand the composition of the microbiome and genetic factors to find appropriate targets for therapeutic interventions.

## 7. Future Directions

It seems crucial to elucidate the nuanced interplay between microbes within the microbiome and between the microbes and the host as well. The microbiome can have protective features for the host, which would favor a definition of mutualistic microbes rather than symbiotic microbes. A large study with 300 patients planned for inclusion was recently initiated with the aim to characterize the microbiome in CTCL.

The skin might not be the only compartment that is relevant for the disease development and course of the disease. Recently, the importance of the composition of the gut microbiome in cancer patients when treated with immunotherapeutic agents was reported [47]. In the future, the examination of the gut microbiome in CTCL patients might reveal interesting aspects with potential implications for treatment and/or diagnosis. According to the previously published data, studies combining the targeting of STAT signaling and *SA* could lead to promising results [48]. STAT3 seems to play a central role in inflammation-mediated cancer [49].

To obtain insights into the pathogenesis, analyses on three-dimensional (3D) human skin culture models might provide informative and tractable experimental systems to examine how *SA*, immune cells, and malignant cells interact and to look into environmental factors which influence the microorganisms colonizing the skin [1]. 

The composition of the skin microbiome could potentially be used as a biomarker, disease-classifier, or treatment target for anti- or probiotic agents or immunomodulatory drugs.

## Figures and Tables

**Figure 1 cancers-14-01324-f001:**
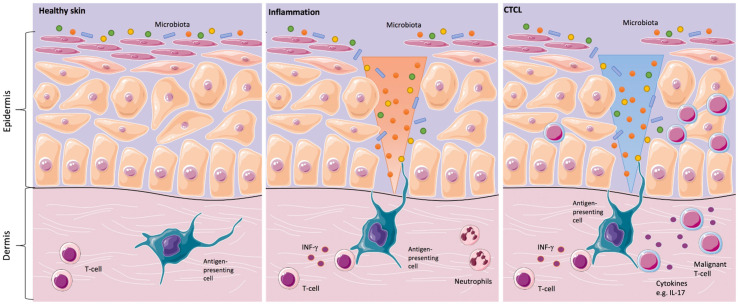
Skins’ interactions with microorganisms in healthy skin, inflammation, and in primary cutaneous T-cell lymphomas (CTCL). Healthy skin in homeostasis (**left image**). Classical inflammatory responses, as characterized by infiltrating neutrophils and monocytes, alongside interferon-γ (IFN-γ)-producing T-cells. In CTCL, a chronic antigen stimulus from interaction with microorganisms is under suspicion in the origin or/and the progression of the disease (**middle image**). For instance, Staphylococcal enterotoxin A can stimulate the activation of signal transducer and activator of transcription 3 (STAT3), as shown in in vitro CTCL models, resulting in an upregulation of interleukin IL-17. IL-17 might indirectly influence tumorigenesis by modulating angiogenesis and inflammation (**right image**) [2,13,21].

**Table 1 cancers-14-01324-t001:** Methods and controls used in recent skin microbiome studies.

Author/Year	Patients	Skin Swabs	16S rRNA Gene Sequencing	Shotgun Metagenomics	Control Skin Swabs	Statistically Significant Differences
Salava et al./2020 [24]	20	lymphoma-affected (MF)	completed	completed	healthy-appearing, contra-lateral	none detected
Salava et al./2017 [25]	13	parapsoriasis-affected	completed	not completed	healthy-appearing, contra-lateral	none detected
Harkins et al./2020 [21]	6	lymphoma-affected (MF/SS)	not completed	completed	healthy volunteer (lower back, thigh)	none detected

**Table 2 cancers-14-01324-t002:** Antiseptic and antibiotic regimen in CTCL therapy.

Author/Year	Antiseptics	Antibiotics	Frequency	Further Strategies
Nguyen et al./2008 [23]	bath with diluted sodium hypochlorite 6%	none	daily to once weekly	none
Lewis et al./2018 [43]	whirlpool bathing system with chlorhexidine gluconate 0.4% rinse with 0.25% acetic acid	i.v. antibiotics (vancomycin and cefepime)	no specification	steroids applied under wet wraps (0.1% Triamcinolone ointment, respectively 1.0–2.5% hydrocortisone cream) Mupirocin 2% ointment alternating with 1% silver sulfadiazine cream on open ulcers followed by skin moisturization (ammonium lactate 12% cream or a glycerin- or lipid-based cream)
Lindahl et al./2019 [22]	none	day 1–10 cephalosporin and metronidazole day 11–24 amoxicillin and clavulanate	once	none
El Sayed et al./2021 [44]	none	doxycycline	daily up to 24 weeks	none

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
