# Peer review of "The Skin Microbiome and Influencing Elements in Cutaneous T-Cell Lymphomas"

_cancers, 2022, doi:10.3390/cancers14051324_

Round 1

Reviewer 1 Report

The present review by Jost & Wehkamp summarizes the current literature about the microbiome and infections in CTCL. The review covers a highly relevant topic: role of the microbiome in the pathogenesis and progression of CTCL. Jost & Wehkamp nicely introduce CTCL and present the studies performed on the microbiome in CTCL patients. The authors describe studies regrading Staphylococcus aureus including how bacterial infection play an important role in the disease progression. They discuss treatment options and the possible advantages of combining cancer-targeting and microbiome-targeting treatments. The review briefly describes some aspects of the skin barrier and its role in CTCL, including reduced filaggrin expression and dysregulated expression of antimicrobial peptides – but mainly in relation to their role in the defense against bacterial infections.

While the topic is important and the review nicely covers an appropriate breadth of literature, I have some concerns that would need to be addressed for the article to be ready for publication:

  1. The review is highly focused on the microbiome and its interplay with the host immune system and malignant CTCL cells, rather than the constituents, composition and interplay of the skin barrier. As such, the title of the review may be more accurate if changed to something related to the skin microbiome rather than the barrier.

  1. The manuscript could greatly benefit from additional proofreading, structure and language editing. Some sentences are very long, making it hard to follow. For instance, in the paragraph line 35-39, the description is vague and hard to read. Section 4 is hard to read due to very punctuated statements and incomplete sentences. Sentence structure and argument is not logical in line 313-315. The structure of the review could be improved. For instance, cutaneous T-cell lymphoma is introduced both in the introduction and in its own section. Some sections are very sparse containing only a single reference to another review and unclear relevance (such as 2.1, 2.3 and 2.4). If you do not include additional references for primary studies but only refer to a single other review, consider stating that these topics are reviewed elsewhere.

  1. Some statements are not supported by the referred study. Line 295 states that barrier dysfunction is likely due to microbiota, yet referred study [18] only describes the barrier defect, not the underlying cause. Some statements are missing references such as line 63-65 (regarding cancer-induced barrier defects). Furthermore, the authors should make sure to cite primary studies when referring to specific results in the text. For example line 176-181, two reviews are cited [32,33] for a sentence starting with “Ødum et al. suggests”, yet none of the references are by Ødum et al. and the actual study is not cited in this context.

  1. In discussion at line 282 and onwards regarding aureus colonization in psoriasis and AD. These statements are lacking clear references and are only covering one view in a controversial topic (Fyhrquist, Muirhead, Prast-Nielsen, Jeanmougin et al. 2019 Nat Commun show a different picture in psoriasis). If included, these statements need to be better supported and possible elaborated to acknowledge the different results and conclusions.

Minor comments and suggestions

  • While it is great to see a schematic figure in the review, the relevance of the included figure is not obvious – especially considering the sparse description in the text and rather vague figure legend. In its current form, Figure 1 does not contribute appreciably to the review.
  • In section 5. It may be worth changing the order of the paragraph to depict the chronology of the studies. The referenced 2008 study lays the foundation for the referred 2018 study and not vice versa.
  • The authors state that the role of Epidermidis is controversial. While the role of S. Epidermidis is definitely complex, there are not obvious controversies. Consider rephrasing.
  • Staphylococcus and its abbreviations should be italic

Reviewer 2 Report

Jost and Wehkamp review the skin barrier in cTCL. The review is well written and structured. However, I would like to point out some concerns regarding the current draft.

Major: 

  1. The review is way to focused on the microbiome (basically Staphylococcae) and MF.
    1. Also other aspects of skin barrier lymphoma interaction should be discussed in more detail as e.g. lymphoma mediated affection of the skin barrier.  
    2. The microbiome is not only comprising S. aureus. How does the microbiome change in cTCL and what is the impact of that change. 
    3. Body of evidence - especially differences to MF - in other cTCL should be included.
    4. Results of TCR clone alignments to e .g bacterial or tissue structures and its shift under treatment should be discussed in more detail.

  Minor: 

  1. References should be checked as e.g. in page 6 line 263. Faclon et al... and refering to two not matching references.

Reviewer 3 Report

The review covers hot topics on recent studies about skin barrier in CTCL and it is comprehensive. The components of this review is very good.

In the section 5 Antiseptic and antibiotic treatment, I thought it would be appreciated if you could make a table summarizing the potential antiseptic and antibiotic treatment which were reported previously like Duvic regimen etc as it would help readers to understand easily what is the common therapeutics of antibiotic treatment for CTCL (I mean not for active bacterial infection like SA in CTCL).

In the section 3 Bacteria, there only two bacteria which this review discussed (SE and SA). Though these two common bacteria are the most important ones, it would be good to add another section like other bacteria. At least, Harkins et al reported that Corynebacterium and Cutibacterium were observed and showed non-significant difference in skin swab sequencing analysis. So, it would be good to add comments on other bacteria even though there is little or no evidence to show that other bacteria are associated with the disease progression of CTCL. 

Minor things

195: skins > skin

Reviewer 4 Report

Jost et al, reported a well conducted review about potential role of microbiome in CTCL, particularly in mycosis fungoides. They analysed three papers in which microbiome have been studied by next generation sequencing techniques in CTCL patients. Although  few studies are still published on this topic and the untill data didn't reach statistical significance, the review is well written and could drive some future studies in understanding  the potential role of microbiome in CTCL in order to find appropriate targets for therapeutic interventions. 

Round 2

Reviewer 1 Report

The authors have addressed most of our concerns and suggestions and we have no further concerns that are critical to address before publication.

Reviewer 2 Report

The authors replied to all remarks made by the reviewer.